# The Ubiquitin Moiety of Ubi1 Is Required for Productive Expression of Ribosomal Protein eL40 in *Saccharomyces cerevisiae*

**DOI:** 10.3390/cells8080850

**Published:** 2019-08-07

**Authors:** Sara Martín-Villanueva, Antonio Fernández-Pevida, Dieter Kressler, Jesús de la Cruz

**Affiliations:** 1Instituto de Biomedicina de Sevilla, Hospital Universitario Virgen del Rocio/CSIC/Universidad de Sevilla, E-41013 Seville, Spain; 2Departamento de Genética, Universidad de Sevilla, E-41012 Seville, Spain; 3Unit of Biochemistry, Department of Biology, University of Fribourg, CH-1700 Fribourg, Switzerland

**Keywords:** ribosome biogenesis, pre-rRNA processing, ribosomal protein L40 (eL40), ubiquitin, UBI1/2 genes, translation, yeast

## Abstract

Ubiquitin is a highly conserved small eukaryotic protein. It is generated by proteolytic cleavage of precursor proteins in which it is fused either to itself, constituting a polyubiquitin precursor of head-to-tail monomers, or as a single N-terminal moiety to ribosomal proteins. Understanding the role of the ubiquitin fused to ribosomal proteins becomes relevant, as these proteins are practically invariably eS31 and eL40 in the different eukaryotes. Herein, we used the amenable yeast *Saccharomyces cerevisiae* to study whether ubiquitin facilitates the expression of the fused eL40 (Ubi1 and Ubi2 precursors) and eS31 (Ubi3 precursor) ribosomal proteins. We have analyzed the phenotypic effects of a genomic *ubi1∆ub*-HA *ubi2∆* mutant, which expresses a ubiquitin-free HA-tagged eL40A protein as the sole source of cellular eL40. This mutant shows a severe slow-growth phenotype, which could be fully suppressed by increased dosage of the *ubi1∆ub*-HA allele, or partially by the replacement of ubiquitin by the ubiquitin-like Smt3 protein. While expression levels of eL40A-HA from *ubi1∆ub*-HA are low, eL40A is produced practically at normal levels from the Smt3-S-eL40A-HA precursor. Finally, we observed enhanced aggregation of eS31-HA when derived from a Ubi3∆ub-HA precursor and reduced aggregation of eL40A-HA when expressed from a Smt3-S-eL40A-HA precursor. We conclude that ubiquitin might serve as a *cis*-acting molecular chaperone that assists in the folding and synthesis of the fused eL40 and eS31 ribosomal proteins.

## 1. Introduction

Ubiquitin is a small eukaryotic protein of 76 amino acids whose name (i.e., it occurs *ubiquitously*) results from its remarkable evolutionary conservation [1]. Ubiquitin functions as a reversible post-translational modifier of proteins to regulate many different cellular processes such as DNA repair, chromatin dynamics, cell cycle regulation, membrane and protein trafficking, endocytosis, autophagy, but most notably proteasome-dependent protein degradation [2,3,4]. Normally, the conjugation of ubiquitin to other proteins involves the formation of an isopeptide bond between the α-carboxyl group of the C-terminal glycine of a ubiquitin molecule with an ε-amino group of a specific lysine residue within the target protein [2,3]. This process is known as ubiquitination and involves three sequential steps: activation, conjugation, and ligation, which are performed by ubiquitin-activating (E1) enzymes, ubiquitin-conjugating (E2) enzymes, and ubiquitin ligases (E3), respectively. The reversal of this modification, deubiquitination, is catalyzed by specific proteases called deubiquitinating enzymes [5].

In most eukaryotes studied, ubiquitin is encoded by two classes of genes: (i) The first comprises genes coding for a single copy of ubiquitin fused to ribosomal proteins (r-proteins), most commonly eL40 and eS31. Ubiquitin fusion to other r-proteins, such as P1 or P2, and even to non-ribosomal proteins such as actin, has been reported in diverse genera of rare single-celled algae as, for example, *Bigelowiella* [6]. (ii) The second class includes genes that encode a polyubiquitin precursor protein, which consists of a polymer of several tandem ubiquitin monomers (e.g., [7,8,9,10]). For both classes, free de novo ubiquitin is synthesized by proteolytic maturation from the corresponding precursor proteins. Only rarely, ubiquitin is encoded by monoubiquitin genes, as occurs in the protozoa *Giardia lamblia* and *Entamoeba histolytica*, or in algae as *Guillardia theta* and *Bigelowiella natans* [11,12,13].

In *Saccharomyces cerevisiae* (thereafter yeast), but also in mammals, ubiquitin is encoded by four different genes [14]. The yeast *UBI4* gene (equivalent to the human *UBB* and *UBC* genes) encodes a polyubiquitin precursor protein of five head-to-tail repeats of ubiquitin. Expression of *UBI4* occurs when cells enter the stationary phase or when they are exposed to diverse stresses, including heat shock, oxidative stress, exposure to DNA-damaging agents, or nutrient depletion [8,15,16,17,18,19]. The paralogous *UBI1* and *UBI2* yeast genes (equivalent to the human *UBA52* gene) code for a fusion protein where a single ubiquitin moiety is fused to the 60S r-proteins eL40A and eL40B, respectively. In turn, the yeast *UBI3* gene (equivalent to the human *RPS27A/UBA80* gene) codes for a fusion protein consisting of a single N-terminal ubiquitin moiety and the 40S r-protein eS31 [8,14]. In actively growing yeast cells, most of the ubiquitin originates from these three r-protein fusion genes [15]; on the other hand, expression of *UBI4* is induced when the *UBI1*, *UBI2*, and *UBI3* genes are turned down, as occurs when cells are subjected to different stresses, including starvation [20].

There are other proteins that share the ubiquitin fold, and these can also be conjugated, akin to ubiquitin, via an isopeptide bond to substrate proteins in an enzymatic process that is analogous to ubiquitination (for a review, see [3,21]). Among these proteins, known as ubiquitin-like modifiers, SUMO (small ubiquitin-related modifier) is the most widely spread member within eukaryotes [21]. Four SUMO genes coding for different monomeric SUMO isoforms have been identified in mammals, while yeast harbors a single essential gene, *SMT3*, which encodes the SUMO protein Smt3 [22]. Sumoylation of target proteins regulates various cellular processes by modulating the localization or activity of the SUMO-modified substrate proteins. While polyubiquitination mainly provides a signal for proteasome-dependent degradation (e.g., [23]), sumoylation seems to enhance the stability of the target proteins (reviewed in [22,24]). This latter feature has been biotechnologically exploited to increase the yield of recombinant proteins in *Escherichia coli*; thus, the N-terminal fusion of a single SUMO moiety to recombinant proteins, which fail to properly fold and precipitate as inclusion bodies, may significantly improve their stability and solubility (e.g., [25,26]). Similarly, expression of recombinant proteins containing an N-terminal ubiquitin moiety has been found to augment their yield and solubility [27,28,29]. However, whether an N-terminally attached SUMO or ubiquitin moiety could act in vivo as a physiological *cis*-acting chaperone for proper folding and efficient expression of the fused proteins remains to be determined (see discussion in [30]).

We are interested in understanding the contribution of the ubiquitin moiety within the Ubi1/2 and Ubi3 precursors to ribosome biogenesis and function. Experimental evidence indicates that while eS31 is a quasi-essential r-protein that assembles in the nucleus, most likely into early 90S pre-ribosomal particles [31,32], eL40 is an essential r-protein that associates in the cytoplasm with late pre-60S r-particles [33,34]. Under wild-type conditions, the Ubi1/2 and Ubi3 precursor proteins could so far never be detected, suggesting that their proteolytic maturation occurs very rapidly, likely co-translationally, and consequently before the assembly of the respective r-proteins into pre-ribosomal particles [14,31,35]. We have previously studied the consequences of introducing mutations in the intersection between ubiquitin and eS31 within the Ubi3 precursor and between ubiquitin and eL40 within the Ubi1 precursor that partially or totally impair ubiquitin removal [35,36]. The obtained results indicate that the presence of ubiquitin hinders the assembly of the respective r-proteins into pre-ribosomal particles; therefore, assembly of either eS31 or eL40A is favored over that of non-cleaved Ubi3 or Ubi1, respectively [35,36]. Moreover, non-cleaved Ubi3 or Ubi1 variants confer lethal phenotypes when expressed as the sole source of eS31 or eL40, respectively. Only in these circumstances (i.e., upon depletion of the wild-type eS31 or eL40 counterparts), these mutant proteins get incorporated into nascent pre-ribosomal particles [35,36]. Strikingly, while Ubi3-containing r-particles can quite efficiently engage in translation [35], Ubi1-containing r-particles are delayed in their cytoplasmic maturation (i.e., Tif6 recycling), prevent r-subunit joining, and interfere with translation elongation [36]. In addition, we and others have shown that the ubiquitin moiety of the Ubi3 precursor can be deleted (*ubi3∆ub* allele) without causing a deleterious phenotype as long as the unfused eS31 r-protein is expressed at an elevated dosage [14,35]. In contrast, expression of eS31 from an *ubi3∆ub* allele integrated into its natural chromosomal locus conferred a pronounced slow-growth (sg) phenotype and a shortage of 40S r-subunits [14,35]. Thus, the N-terminal ubiquitin moiety of Ubi3 contributes to the efficient synthesis of eS31 and, as a consequence, is required for the quantitative synthesis of 40S r-subunits. In this study, we have performed an equivalent analysis with a *ubi1∆ub* allele, which encodes a ubiquitin-free Ubi1∆ub protein. As above, when the sole cellular source of eL40 originates from a single-copy *ubi1∆ub* allele, integrated at the native *UBI1* genomic locus, cells showed a pronounced sg phenotype and a deficit in 60S r-subunits; both defects are fully suppressed by increasing the gene dosage of the *ubi1∆ub* allele by placing it on a centromeric (*CEN*) plasmid. Therefore, as for Ubi3, it appears that the ubiquitin moiety of the Ubi1 precursor is important for the efficient accumulation of the derived eL40A r-protein product. Importantly, expression of an N-terminally Smt3-fused eL40A variant protein as sole source of eL40 leads to a modest yet significant improvement of cell growth, especially at high temperatures. Finally, we have evaluated the effect of the absence and presence of an N-terminal ubiquitin or Smt3 moiety, respectively, on the aggregation status of HA-tagged eL40A, either expressed from a genomic *UBI1*-HA, *ubi1∆ub*-HA or *SMT3-S-eL40A*-HA allele, and HA-tagged eS31, expressed either from a genomic *UBI3*-HA or *ubi3∆ub*-HA allele. While we found practically no differences in the extent of aggregation for HA-tagged eL40A derived from the ubiquitin-containing Ubi1-HA precursor or the ubiquitin-free Ubi1∆ub-HA protein in cells encompassing a *ubi2∆* mutation, we observed both a clear tendency for reduced aggregation of eL40A-HA when generated from the Smt3-S-eL40A-HA precursor and for enhanced aggregation of HA-tagged eS31 derived from the *ubi3∆ub*-HA allele when compared to eS31-HA originating from the *UBI3-HA* allele. We discuss our results in the context of a possible *cis*-acting role of the N-terminal ubiquitin moiety, fused to eS31 and eL40, as a molecular chaperone that facilitates the correct folding and efficient synthesis of these two r-proteins.

## 2. Materials and Methods

### 2.1. Strains and Microbiological Methods

All yeast strains used in this study are listed in Table 1 and are derived from the diploid W303 strain [37]. A two-step allele replacement method was used to replace genomic wild-type *UBI1* by the *UBI1*-HA, *ubi1∆ub*-HA or *SMT3-S-eL40*-HA alleles [38]. Briefly, SMY106, which is a haploid *ubi1*::klURA3 (*URA3* gene from *Kluyveromyces lactis*) null mutant strain, was co-transformed with 100 ng of empty YCplac111 vector and 1 µg of DNA fragments containing the *UBI1*-HA, *ubi1∆ub*-HA, or *SMT3*-S-eL40-HA alleles excised by *Eco*RI/*Hin*dIII digestion from YCplac111-UBI1-HA, YCplac111-*ubi1∆ub*-HA, or YCplac111-SMT3-S-eL40-HA, respectively. Transformants were selected on SD-Leu plates. Then, transformants were replica-plated onto 5-FOA-containing plates to select for those that had lost the klURA3 marker as a consequence of a site-specific recombination event. Candidate clones were analyzed by colony PCR and sequencing. SMY113 (*UBI1*-HA), SMY107 (*ubi1∆ub*-HA), and SMY218 (*SMT3*-S-*eL40A*-HA) are representative 5-FOA-resistant clones that were selected for further analyses. To obtain strains where the above alleles are the sole source of cellular eL40, SMY113, SMY107, and SMY218 were crossed to JDY923, which is a *ubi2*::kanMX4 null mutant. The resulting diploids were sporulated and tetrads were dissected. Several complete tetrads from each cross were analyzed for (i) G418 resistance on G418-containing plates, (ii) the presence of the *ubi2*::kanMX4 deletion disruption and either the *UBI1*-HA, *ubi1∆ub*-HA or *SMT3*-S-*eL40A*-HA allele by PCR, and (iii) the expression of the HA epitope by immunoblotting using anti-HA antibodies. SMY256 (*UBI1*-HA *ubi2∆*), SMY257 (*ubi1∆ub*-HA *ubi2∆*), and SMY258 (*SMT3*-S-*eL40A*-HA *ubi2∆*) are representative meiotic segregants that were selected for further analyses. Strains SMY215 and SMY216 were generated by transforming the *UBI1/2* shuffle strain (TAY001) with the plasmids YCplac111-ubi1∆ub-HA or YCplac111-SMT3-S-eL40-HA; transformants were then restreaked on 5-FOA-containing plates to counter-select against the presence of pHT4467∆-UBI1.

Yeast cells were grown at the indicated temperatures either in YPD medium (1% yeast extract, 2% peptone, and 2% glucose) or SD medium (synthetic dextrose; 0.15% yeast nitrogen base, 0.5% ammonium sulphate, and 2% glucose), which was supplemented with the appropriate amino acids and bases as nutritional requirements. To prepare plates, 2% agar was added to the media before sterilization. Yeast genetic techniques and growth media have been previously described [39]. Yeast cells were transformed by the lithium acetate method [40]. For tetrad dissection, a Singer MSM 400 micromanipulator was used. Standard molecular biology techniques were carried out according to the specifications in Sambrook et al. [41]. *E. coli* DH5α was used for cloning and propagation of plasmids [41].

### 2.2. Plasmids

Plasmids used in this study were generated by multiple sequential cloning steps and they are listed in Table 2. All constructs were verified by DNA sequencing. Description of the oligonucleotides used for the PCRs and information on the construction of the different plasmids will be available upon request.

### 2.3. Polysome Analysis and Sucrose Gradient Fractionation

Cell extracts for polysome analysis were prepared and analyzed as previously described [44,45]. Ten A_260_ units of each cell extract were loaded onto 7–50% sucrose gradients. These gradients were centrifuged at 39,000 rpm in a Beckman Coulter SW41Ti rotor for 2 h 45 min; the A_254_ was continuously monitored using an ISCO UA-6 system. When needed, fractions of 0.5 mL were collected and proteins were precipitated from each fraction using trichloroacetic acid, as previously described [46]. The precipitated fractions were resuspended in 2× Laemmli loading buffer; an equal volume of each fraction was separated by SDS-PAGE and analyzed by western blotting.

### 2.4. Western Blot Analysis and Antibodies

Total yeast protein extracts were prepared by the alkaline lysis method of Yaffe and Schatz [47], which immediately *freezes* the in-vivo protein content, thus preventing rapid protein turnover reactions. Proteins were separated by SDS-PAGE and analyzed by western blotting according to standard procedures [41]. The following primary antibodies were used: mouse monoclonal anti-HA 16B12 (Covance, Emeryville, CA, USA), anti-Pgk1 (Invitrogen, Eugene, OR, USA), and anti-uL3 (gift from J.R. Warner) [48]; rabbit polyclonal anti-uL29 (gift from M. Seedorf) [49] and anti-uS3 (gift from M. Seedorf) [49]. Goat anti-mouse or anti-rabbit horseradish peroxidase-conjugated antibodies (Bio-Rad, Hercules, CA, USA) were used as secondary antibodies. Immune complexes were visualized using a chemiluminescence detection kit (Super-Signal West Pico, Pierce, Rockford, IL, USA) and a ChemiDoc^TM^ MP imaging system (Bio-Rad, Hercules, CA, USA).

### 2.5. Analysis of Aggregated Proteins

Isolation of aggregated proteins was done from whole cell extracts as described by Koplin et al. [50], but following the sonication modifications described in Panasenko et al. [51]. Briefly, 40 OD_600_ units of cells, grown to mid-log phase in YPD, were harvested, washed with water containing 15 mM NaN_3_, and cell pellets were first frozen in liquid N_2_ and then stored at −80 °C. For preparation of cell lysates, the pellets were resuspended in 0.5 mL of freshly prepared lysis buffer (20 mM sodium phosphate, pH 6.8, 10 mM DTT, 1 mM EDTA, 0.1% Tween 20) supplemented with 1× Complete EDTA-free protease inhibitor cocktail (Roche, Basel, Switzerland), 3 mg/mL zymolyase T20 (USBiological, Salem, MA, USA) and 1.25 U/mL benzonase. Samples were incubated at room temperature for 15 min, chilled on ice for 5 min, and tip sonicated (Branson sonifier 450; three times 10 s at duty cycle 40%). Upon sonication, the lysates were cleared by centrifugation at 200× *g* for 20 min at 4 °C, and the supernatants were adjusted to an identical protein concentration, determined by the Bradford method, of 10.0 mg/mL. Supernatant aliquots of 20 µL were taken and boiled with 40 µL of SDS Laemmli sample buffer (total extract samples). Then, the supernatants were centrifuged at 16,000× *g* for 20 min at 4 °C to pellet the aggregated proteins. After removing the supernatants, the pellets were washed twice with 0.5 mL of wash buffer (20 mM sodium phosphate, pH 6.8, 1× Complete EDTA-free protease inhibitor cocktail) containing 2% NP-40 (WB-NP); then, pellets were resuspended in 0.5 mL WB-NP, sonicated (10 s at duty cycle 40%), and centrifuged at 16,000× *g* for 20 min at 4 °C. Pellets were again washed twice with 0.5 mL of wash buffer and then resuspended in wash buffer, sonicated, and centrifuged as above. Pellets were resuspended in 50 µL of SDS Laemmli sample buffer and boiled (protein aggregate samples). Total extract and protein aggregate samples were then separated by 4–12% gradient SDS-PAGE and analyzed by colloidal blue Coomassie staining and western blotting.

### 2.6. Reproducibility

To ensure good reproducibility, each experiment was done two or more times (biological replicates); then, each sample was also analyzed at least twice (technical replicates). Representative experiments are shown in the different figures.

## 3. Results

### 3.1. Generation of ubi1 Mutants for Phenotypic Analysis

Yeast eL40 is an essential r-protein of 52 amino acids that is encoded by two independent paralogous genes, *UBI1* (*RPL40A*; YIL148W), and *UBI2* (*RPL40B*; YKR094C). As mentioned in the introduction, eL40 as well as the r-protein eS31 (*UBI3* gene) are produced in practically all eukaryotes by proteolytic removal of the N-terminal ubiquitin moiety from the ubiquitin-fused precursor r-proteins. We previously studied the effects of impaired ubiquitin removal from the Ubi1 and Ubi3 precursors by introducing specific mutations that decrease or abolish their proteolytic maturation [35,36].

To date, however, only experimental data concerning the phenotypic consequences of expressing a genomic *ubi3∆ub* allele, which lacks the region coding for the ubiquitin moiety, are available [14,35]. To explore the role of the N-terminal fused ubiquitin moiety within Ubi1/2 precursors, we constructed two different mutant Ubi1 variants (Figure 1). First, we replaced the endogenous wild-type *UBI1* copy with a *ubi1∆ub*-HA allele, which harbors a deletion of the ubiquitin coding region and therefore solely expresses the eL40A r-protein part without the N-terminal ubiquitin moiety. To allow efficient translation of the construct, the cognate ATG start codon of *UBI1* was kept downstream of its 5′ untranslated region; thus, the corresponding eL40A protein starts with a N-terminal methionine that is not present in the natural, proteolytically processed eL40A. Second, we replaced the endogenous *UBI1* allele with a *SMT3-S-eL40A*-HA allele where the ubiquitin coding region was exchanged with a Smt3 (hereafter SUMO) coding region. To facilitate cleavage of the SUMO moiety, we inserted a serine codon at the junction between the coding regions of Smt3 and eL40A. In both constructs, as well as in the genomic *UBI1*-HA wild-type control construct, we also added an in-frame C-terminal single HA epitope for western detection.

### 3.2. The ubi1∆ub Ubiquitin Deletion Mutant Displays a Slow-Growth Phenotype

For the functional analysis of the generated mutants, we first examined the growth phenotype associated with the deletion of the ubiquitin moiety from the Ubi1 precursor (Figure 2). As eL40 is encoded by two independent paralogous genes, we performed this analysis both in the presence or the absence of a functional eL40B r-protein (wild-type *UBI2* or *ubi2∆* background, respectively). As previously described [33,36], the presence of the HA epitope slightly affected growth when eL40A-HA is the sole source of eL40, specially at low temperatures. In agreement, doubling times of ca. 1.8 h, 2.0 h, and 3.0 h were obtained in liquid YPD at 30 °C for the wild-type W303-1A strain and the isogenic *ubi2∆* and *UBI1*-HA *ubi2∆* mutants, respectively. As also discussed [36], this is likely due to an adverse effect of the extra amino acids on the interaction environment of the C-terminus of eL40A within the ribosome. Figure 2 also shows that deletion of the ubiquitin moiety has practically no effect on cell growth when the *UBI2* gene is present. However, *ubi1∆ub*-HA *ubi2∆* mutant cells, in which all cellular eL40 is derived from the single-copy expression of a ubiquitin-free eL40A construct, showed a severe sg phenotype at all tested temperatures and divided every 5.5 h in liquid YPD at 30 °C.

It has been previously shown that increased dosage of the *ubi3∆ub* allele suppresses its growth defect [14,35]. Thus, to address whether a similar situation occurred upon increasing the dosage of the *ubi1∆ub* allele, we cloned either the wild-type *UBI1*-HA or the mutant *ubi1∆ub*-HA allele into YCplac111, a centromeric (*CEN*) low-copy-number *LEU2* plasmid [43], and transformed these constructs into the *UBI1/2* shuffle strain TAY001 (for its genotype, see Table 1). After counter-selection against the pHT4467∆-UBI1 (*URA3 ADE3*) plasmid on 5-FOA-containing plates, cell growth was analyzed on YPD plates. As shown in Figure 2, cells harboring the *ubi1∆ub*-HA allele on the *CEN* plasmid, thus constituting the sole source of cellular eL40, grew significantly better than the *ubi1∆ub-HA ubi2∆* mutant cells, in which the *ubi1∆ub-HA* allele is present at the genomic *UBI1* locus. This is in agreement with previous data showing that the expression of a *ubi1∆ub* allele from a strong promoter also complemented the lethal phenotype of a *ubi1∆ ubi2∆* strain to the wild-type extent [14]. We conclude that the presence of the ubiquitin moiety is required for optimal cell growth when eL40 is expressed from a single-copy allele.

### 3.3. The SMT3-S-eL40A ubi2∆ Mutant Displays a Slow-Growth Phenotype

The small ubiquitin-like modifier SUMO has an overall three-dimensional structure that is very similar to the one of ubiquitin and possesses, as ubiquitin, a C-terminal glycine residue via which it is conjugated to specific lysine residues of target proteins (e.g., [52]). Thus, given that ubiquitin provides a beneficial *cis*-acting effect when fused to eL40, we reasoned that the replacement of the ubiquitin moiety in the Ubi1 precursor by SUMO should therefore also permit, if enabling proper processing of the Smt3-eL40A fusion protein by the SUMO proteases Ulp1 and Ulp2 [53], the optimal production of functional eL40A protein. To this end, we integrated the *SMT3-S-eL40A*-HA allele at the genomic *UBI1* locus (see Section 3.1 and Materials and Methods). Then, we assessed the growth of cells harboring the *SMT3-S-eL40A*-HA allele in the context of the wild-type *UBI2* and the *ubi2∆* null mutant background. As shown in Figure 3, *SMT3-S-eL40A*-HA cells grow practically like the wild-type control strain in the presence of *UBI2*. Interestingly, the *SMT3*-S*-eL40A*-HA *ubi2∆* strain, where the Smt3-S-eL40A-HA precursor is the sole source of eL40, displays a mildly improved growth compared to that of the isogenic *ubi1∆ub*-HA *ubi2∆* strain, which is especially evident at higher temperatures. Consistently, when grown in liquid YPD at 30 °C, *SMT3*-S*-eL40A*-HA *ubi2∆* cells divided more rapidly than *ubi1∆ub*-HA *ubi2∆* cells as indicated by the determined doubling times of 4.0 h and 5.5 h, respectively. As observed for the *ubi1∆ub*-HA allele, introducing the *SMT3*-S*-eL40A*-HA allele on a low-copy-number plasmid into *ubi1∆ ubi2∆* cells resulted in apparent wild-type growth, showing that its increased gene dosage suppressed the sg phenotype linked to the genomic single-copy *SMT3*-S*-eL40A* allele. We conclude that SUMO is able to partially fulfil the *cis*-acting role of ubiquitin within the Ubi1 precursor when replacing it as the N-terminal fusion moiety preceding the eL40A r-protein component.

### 3.4. Contribution of the Ubiquitin and SUMO Moieties to the Expression of eL40A

The fact that the sg phenotype linked to providing either the *ubi1∆ub*-HA or the *SMT3*-S*-eL40A*-HA allele as the sole source of cellular eL40 r-protein was suppressed by increasing the copy number of these alleles strongly suggests that the initial presence of the ubiquitin moiety within the Ubi1 precursor ensures that the released eL40A r-protein is produced in sufficient amounts. To test this hypothesis, we monitored the expression levels of the eL40A-HA r-protein in the different used strains by western blotting. As shown in Figure 4, the protein levels of eL40A-HA were significantly lower in the *ubi1∆ub*-HA *ubi2∆* strain when compared with those of the *UBI1*-HA *ubi2∆* strain. As expected, an increased dosage of the *ubi1∆ub*-HA allele fully restores the levels of eL40A-HA (Figure 4, lane 5). Together, these results indicate that the ubiquitin moiety of Ubi1 is required for the efficient production of the eL40 r-protein.

To assess the efficiency of cleavage of the Smt3-S-eL40A-HA precursor into SUMO and eL40A-HA, western blot analyses using anti-HA antibodies were also carried out with whole cell extracts of strains harboring a genomic *SMT3*-S*-eL40A*-HA allele. As previously shown [33,36], no Ubi1-HA precursor but only mature eL40A-HA was detected in the *UBI1*-HA strains (Figure 4, lanes 2 and 3). In contrast, although cell extracts of the *SMT3*-S*-eL40A*-HA *ubi2∆* strain mostly contained mature eL40A-HA, some precursor protein could also be detected at the expected molecular mass, especially when the Smt3-S-L40A-HA fusion protein variant was the sole cellular source of eL40A-HA (Figure 4, lanes 6 and 7). We conclude that proteolytic processing of the Smt3-S-L40A-HA precursor into SUMO and eL40-HA is less efficient than removal of ubiquitin from the Ubi1 precursor. Nevertheless, we observed a clear increase in the total amount of produced mature eL40A-HA in the cell extracts of the *SMT3*-S*-eL40A*-HA *ubi2∆* strain when compared to the eL40A-HA levels present in the *ubi1∆ub*-HA *ubi2∆* strain.

### 3.5. The Genomic ubi1∆ub-HA and SMT3-S-eL40A-HA Alleles Affect the Functional Integrity of 60S r-Subunits

As the genomic *ubi1∆ub*-HA allele limits the production of the eL40A protein in a *ubi2∆* background (Figure 4), we wondered whether this would affect the synthesis and function of 60S r-subunits. To answer this question, we performed polysome profile analyses with cell extracts obtained from this strain and, to evaluate the effect of the SUMO fusion, the *SMT3*-S*-eL40A*-HA *ubi2∆* strain. First, we examined the profiles of the *UBI1*-HA *ubi2∆* control strain grown in YPD at 30 °C. This strain, in agreement with its slightly slower growth (Figure 2), still displayed good amounts of actively translating ribosomes, as inferred from the overall polysome content (Figure 5A). However, compared to *ubi2∆* mutant cells for which we had previously reported a mild deficit in free 60S r-subunits [33], the levels of free 60S r-subunits were rather moderately increased and half-mer polysomes could be observed, suggesting that there is some impairment of subunit joining when the Ubi1-HA precursor is expressed from the genomic locus in *ubi2∆* cells. In contrast, the analysis of the *ubi1∆ub*-HA *ubi2∆* strain revealed, in addition to the appearance of half-mer polysomes, a drastic decrease in polysome content (Figure 5B). Notably, this profile closely resembles those obtained upon the depletion of eL40, including the accumulation of both r-subunits in their free forms [33], thus strongly suggesting that the amount of available eL40 is indeed limited in this strain. Consistently, increasing the copy number of the *ubi1∆ub*-HA allele, by providing it on a centromeric plasmid in the context of the *ubi1∆ ubi2∆* background, resulted in a wild-type polysome profile, as this overcame the limitation in eL40 availability (Figure 4 and Figure 5C). Altogether, these results indicate that the deletion of the ubiquitin moiety from a genomically expressed ubiquitin-free Ubi1∆ub protein, in the absence of eL40B, leads to a decreased production of eL40A and as a consequence affects the synthesis and function of 60S r-subunits. Thus, we conclude that, as also reported to be the case for the ubiquitin moiety of Ubi3 [14,35], the ubiquitin moiety of Ubi1 is important but not strictly required for the production of eL40A and hence the synthesis of 60S r-subunits.

We also assessed the effects of replacing ubiquitin by the SUMO moiety. As shown in Figure 5D, the *SMT3*-S*-eL40A*-HA *ubi2∆* strain displayed a polysome profile with reduced polysome content and, more notably, with a clear increase in the levels of free 60S r-subunits *versus* free 40S r-subunits that was accompanied by the occurrence of half-mer polysomes; such a profile is typical for mutants with a pronounced defect in 60S to 40S r-subunit joining (e.g., [54,55,56]). Strikingly, we have previously observed highly similar polysome profiles, albeit with varying degrees of polysome content, in cells expressing mutant Ubi1 precursors that are not efficiently proteolytically processed into ubiquitin and eL40A [36], suggesting that the accumulating free 60S r-subunits may contain the non-cleaved Smt3-S-eL40A-HA protein.

Next, we addressed how the mature eL40-HA or the non-cleaved precursors, derived from Ubi1-HA, Ubi1∆ub-HA, and Smt3-S-eL40A-HA, were distributed along the different ribosomal particles (i.e., free 60S r-subunits, vacant 80S couples/monosomes, and translating ribosomes) by sucrose gradient fractionation. No apparent differences were found between the distribution of eL40-HA expressed either from the *ubi1∆ub*-HA or the *UBI1*-HA allele in *ubi2∆* null mutant cells. In both cases, eL40A-HA was, similarly to the 60S r-subunit control protein uL3, detected in the 80S ribosome and polysome fractions. This result indicates that the fused ubiquitin moiety is not required for the assembly of the r-protein eL40A protein into functional 60S r-subunits (Figure 6, upper and middle panel). In contrast, when the *SMT3-S-eL40A*-HA *ubi2∆* strain was studied, the mature eL40A-HA fractionated similarly as the 60S r-subunit control protein uL3; however, the non-cleaved Smt3-S-eL40A-HA mainly peaked in the 60S and 80S fractions (Figure 6, bottom panel). Notably, this precursor protein was almost completely absent from the polysomal fractions. This result, which is similar to what we have previously observed for non-cleaved Ubi1 precursor variants [36], indicates that the Smt3-S-eL40A-HA precursor can assemble into pre-60S r-particles that are, however, unable to efficiently engage in subunit joining and translation elongation, suggesting that the resulting 80S ribosomes are not competent to enter the pool of translating ribosomes. Strikingly, uL3 but not the mature eL40A was identified in the 60S peak (see Figure 6, bottom panel, lanes 7–10); this result further supports the hypothesis that the 60S r-particles containing eL40A-HA are very efficiently recruited into polysomes while those containing Smt3-S-eL40A-HA are not and accumulate in the 80S area of the gradient (see profile in Figure 5D).

### 3.6. In Trans Expression of Free Ubiquitin Ffails to Rescue the Deficiencies of the ubi1∆ub-HA ubi2∆ and SMT3-S-eL40A-HA ubi2∆ Mutants

To formally exclude the possibility that a ubiquitin shortage might account for the growth defects of *ubi1∆ub*-HA *ubi2∆* and *SMT3-S-eL40A*-HA *ubi2∆* mutant cells, considering that the *UBI3* gene remains the only source of cellular ubiquitin during active growth of these cells, we transformed these two strains, as well as the *UBI1*-HA *ubi2∆* control strain, with a plasmid expressing the ubiquitin moiety to increase the available amount of free ubiquitin or with an empty control plasmid. In case that ubiquitin was indeed limiting, then ectopic *in trans* expression of free ubiquitin would be expected to result in a suppression of the sg phenotype of these mutant cells. As shown in Figure 7A, expressing free ubiquitin *in trans* from a plasmid did, however, not improve the growth of *UBI1*-HA *ubi2∆, ubi1∆ub*-HA *ubi2∆*, and *SMT3-S-eL40A*-HA *ubi2∆* cells. In addition, we performed polysome profile analyses with cell extracts derived from these strains. As expected from the unchanged growth, the additional expression of free ubiquitin did not restore the perturbed synthesis and function of 60S r-subunits in either *ubi1∆ub*-HA *ubi2∆* and *SMT3-S-eL40A*-HA *ubi2∆* cells or in the isogenic *UBI1*-HA *ubi2∆* control strain (Figure 7B and data not shown). Altogether, these results indicate that the contribution of ubiquitin to the synthesis and assembly of eL40 is exclusively exerted when it is fused *in cis* to the r-protein tail but not when it is expressed *in trans* as a free molecule.

### 3.7. Ubiquitin and SUMO Modestly Prevent eL40 or eS31 Protein Aggregation

It has been proposed that the N-terminal ubiquitin moiety present in the Ubi1/2 and Ubi3 precursors acts as a *cis*-acting chaperone that facilitates the correct folding and, hence, the efficient synthesis and assembly of eL40 and eS31, the respective r-protein tails, into pre-ribosomal particles [14,30,35]. However, only genetic evidence in support of this conjecture has so far been reported ([14,35], this study). Thus, to obtain more direct experimental insight into this suspected role of ubiquitin and to explore whether SUMO could act similarly, we monitored the levels of aggregated proteins in cells expressing the different Ubi1 variants used in this study. As a positive control for the protein aggregation assay, we included a strain lacking the Hsp70 chaperone Ssb (a *ssb1∆ ssb2∆* mutant), which has been described to exhibit a pronounced aggregation of a variety of newly synthesized proteins, especially r-proteins and factors involved in r-subunit biogenesis or translation [50]. As shown in Figure 8, substantial protein aggregation was found in strains expressing eL40A-HA as the sole cellular source of eL40 from the genomic *UBI1*-HA (lane 5), *ubi1∆ub*-HA (lane 6) or *SMT3-S-eL40A*-HA (lane 7) alleles, which could not be simply explained by the common presence of the *ubi2∆* null allele. Thus, the presence of the C-terminal HA-tag on eL40 appears to negatively interfere with the synthesis and proper functioning of 60S r-subunits and, hence, besides having some impact on optimal cell growth (see Figure 2 and Ref. [36]), may also promote some protein aggregation (see also below). Moreover, the global aggregation tendency was very similar in the three strains, albeit slightly more pronounced in *ubi1∆ub*-HA *ubi2∆* cells (Figure 8, top panel, lane 6). Western blotting against the HA-tag indicated that eL40A-HA is among the different aggregated proteins in extracts of the above strains, as also are other r-proteins of the large and small r-subunit, such as uL29 and uS3 (Figure 8, lower panels). The housekeeping Pgk1 enzyme, which has so far not been implicated in any aspect of ribosome metabolism, has, however, only a low propensity for aggregation in all tested strains. Strikingly, the levels of aggregated eL40A-HA, although apparently not increasing when eL40A-HA is synthesized from the *ubi1∆ub*-HA allele, clearly diminished when derived from the Smt3-S-eL40A-HA precursor (compare lanes 6 and 7). This result suggests a minor but positive role of SUMO as a *cis*-acting chaperone for proper folding and, thus, solubility of the eL40 r-protein.

We also assessed the levels of aggregated proteins in a *ubi3∆ub*-HA strain and its isogenic *UBI3*-HA control strain. As shown in Figure 9, the aggregation analysis suggested less global accumulation of insoluble proteins in cells of these strains compared to the equivalent *UBI1*-HA *ubi2∆* and *ubi1∆ub*-HA *ubi2∆* strains, which is especially evident for the *UBI3*-HA *versus* the *UBI1*-HA *ubi2∆* strain. Moreover, while western blotting showed practically no aggregation of eS31-HA when produced from the wild-type Ubi3-HA precursor, a clear aggregation of this r-protein was observed when it is produced from the ubiquitin-free Ubi3∆ub protein (Figure 9, lanes 4 and 5), suggesting that the initial presence of ubiquitin may promote the soluble expression of eS31. In addition, the aggregation analyses revealed three other remarkable findings that are worth being mentioned: (i) In all samples containing aggregated eL40A-HA or eS31-HA, also other r-proteins, such as uL29 and uS3, were among the aggregated proteins (see Figure 8 and Figure 9, lower panels). (ii) Aggregation of these r-proteins occurs not only in the *ssb1∆ ssb2∆* control strain, but also in mutant strains lacking a specific r-protein and/or displaying a defect in ribosome biogenesis, as shown here for the *ubi3∆, rps12∆, rpl39∆*, and *dob1-1* mutant. This is consistent with the recent observation that perturbing the assembly of ribosomes leads to the aggregation of newly synthesized r-proteins, which provokes a collapse of overall proteostasis and severely compromises cell growth [57]. (iii) In contrast to what has been previously reported [50], we did find substantial amounts of eS31-HA among the different r-proteins present in the aggregates of *ssb1∆ ssb2∆* cells. Thus, the ubiquitin moiety fused to eS31 appears to be insufficient to protect eS31-HA from aggregation in these circumstances, indicating that this r-protein is also a client of the Ssb chaperone system. Altogether, our observations suggest that although the ubiquitin moiety fused to eS31 and eL40 has an important function for the optimal production of the respective r-protein, its presence does not specifically prevent the aggregation of eS31 when cells lack the Ssb chaperone system. Moreover, they provide further evidence for the notion that defects in ribosome assembly elicit a proteostatic stress that results in the aggregation of r-proteins and, as a consequence, many other proteins.

## 4. Discussion

Ubiquitin is an extremely conserved post-translational modifier that is present in all eukaryotes and is involved in various cellular processes [2,3,4]. In practically all eukaryotes where the organization of the genes encoding ubiquitin has been analyzed, a fusion of ubiquitin with the r-proteins eL40 (Ubi1 and Ubi2 precursors) or eS31 (Ubi3 precursor) is observed [6]. As these particular combinations have been strictly maintained during evolution, it can be assumed that they might be beneficial for the fitness of eukaryotic cells. While fusing the ubiquitin moiety to an r-protein of each r-subunit could be an excellent way in which to couple the synthesis and degradation of proteins in eukaryotes, its explicit fusion to eL40 and eS31 goes beyond this possible strategy and suggests a specific function of ubiquitin with respect to these r-proteins, or vice versa, a specific effect of these two r-proteins on the fused ubiquitin. To understand this puzzling situation, we have previously analyzed the consequences of impairing ubiquitin removal from the Ubi1 and Ubi3 precursors and concluded that these precursors are likely already co-translationally cleaved, with their cleavage being a pre-requisite for the efficient assembly of the two released r-protein tails and their correct function within the r-subunits [35,36]. Whether processing of UBA52 and UBA80, the ubiquitin-eL40 and ubiquitin-eS31 precursor, respectively, in human cells, is co- or post-translational remains to be determined; however, it has recently been suggested, based on results obtained with a reticulocyte lysate-based translation system, which does not exactly reflect the in-vivo situation, that processing of UBA52 can occur post-translationally [58]. In any case, processing of UBA52 has also been shown to be critical for the function of eL40 in human cells [59]. Moreover, we and others have also experimentally addressed the effects of deleting the ubiquitin moiety from the Ubi3 precursor [14,35]; thus, revealing that cells harboring a genomically integrated *ubi3∆ub*-HA allele display a severe sg phenotype [35]. Our study indicated that this growth defect was likely due to the reduced expression of HA-tagged eS31, resulting in a net 40S r-subunit shortage; accordingly, increasing the gene dosage of this allele restored wild-type growth [35]. In this work, we have tackled the analysis of an equivalent genomic *ubi1∆ub*-HA allele. As eL40 is encoded by two paralogous genes in yeast [33], we were forced to study this allele in the context of a *ubi2∆* deletion background in order to achieve that the sole cellular source of eL40A-HA originates from the ubiquitin-free allele. Akin to the *ubi3∆ub*-HA mutant, *ubi1∆ub*-HA *ubi2∆* cells showed, compared to the isogenic *UBI1*-HA strain, a severe sg phenotype and reduced expression levels of eL40A**-HA (Figure 2 and Figure 4), which affects the synthesis and function of 60S r-subunits such that the overall translational activity is clearly decreased (Figure 5). Again, as previously observed for the *ubi3∆ub*-HA allele, increasing the dosage of the *ubi1∆ub*-HA allele fully suppressed all of the above defects (Figure 2 and Figure 5). Thus, our experiments clearly show that the ubiquitin moiety of Ubi1, as it is also the case for Ubi3, is mainly required to facilitate the expression of the eL40 r-protein. The fact that expression *in trans* of free ubiquitin fails to restore the defects of *ubi1∆ub*-HA *ubi2∆* cells (Figure 7) indicates that ubiquitin only exerts its beneficial role when fused in *cis* to eL40A. Moreover, we also concluded from this experiment that the defects associated with the *ubi1∆ub*-HA *ubi2∆* allele were not due to a ubiquitin limitation; indeed, *ubi1∆, ubi2∆*, and *ubi3∆* cells have practically similar levels of ubiquitin as wild-type cells [14].

How does the fused ubiquitin moiety facilitate the expression of the eL40 and eS31 r-protein tails? This is still unclear, and some non-mutually exclusive possibilities, which must also agree with the evolutionary conservation, can be envisaged: (i) It could be that ubiquitin directly assists the assembly of eL40 and eS31. Thus, in the absence of ubiquitin, their assembly into the respective pre-ribosomal particles would be less productive and, therefore, the unassembled r-protein fraction of eL40 or eS31 would be efficiently and rapidly degraded, as has been described to be the general case for r-proteins that are produced in excess and/or fail to properly assemble [60,61,62,63,64]. This possible direct role of ubiquitin is questionable since cleavage of the ubiquitin fusion precursors seems to occur very rapidly, possibly even co-translationally. While it would still be conceptually possible that ubiquitin assists eL40 assembly since this occurs in the cytoplasm [33], it is unlikely that ubiquitin protects eS31 until its assembly into pre-ribosomal particles in the nucleolus [31,32]. Still, in human cells, ubiquitin cleaved from UBA52 has been suggested to form a complex with eL40 for regulation of protein synthesis [59]; however, this complex has not been shown to facilitate eL40 assembly. (ii) A more plausible scenario proposes that ubiquitin fulfils a role as a *cis*-acting chaperone to facilitate translation and folding of eL40 and eS31. While only the correctly folded r-proteins are competent for assembly into pre-ribosomal particles, the misfolded fraction of these will have an increased propensity for aggregation and/or be cleared by degradation. In agreement with such a *cis*-acting role of ubiquitin, we show that the expression of ubiquitin-free eL40A is clearly reduced (Figure 4) and that the replacement of the ubiquitin moiety of Ubi1 by the ubiquitin-like protein SUMO (yeast Smt3), which is known to increase the productive expression of fused proteins [27,28,29,65], is able to partially suppress the growth defect of the *ubi1∆ub*-HA *ubi2∆* mutant, especially at high temperatures (Figure 3), and leads to higher yields of the mature eL40A-HA protein (Figure 4). The fact that the Smt3-S-eL40A-HA precursor is not efficiently processed (Figure 4 and Figure 6) could be the reason for the incomplete suppression, as ubiquitin release from eL40 is required for the cytoplasmic maturation and proper function of 60S r-subunits [36] (see also Figure 5 and Figure 6). Additional experiments are required to understand the reasons why processing of the Smt3-S-eL40A-HA precursor is inefficient, although it could be that the two SUMO proteases Ulp1 and Ulp2, which are presumed to cleave this fusion protein, might be limiting as they are localized in the nuclear periphery and in the nucleus, respectively (reviewed in [53]). Moreover, to further explore the role of ubiquitin as a chaperone of r-proteins eL40 and eS31, we have analyzed the aggregation status of different eL40 and eS31 protein variants. In general, most r-proteins have a high tendency to aggregate during or after their synthesis due to their particular characteristics, such as an extremely high isoelectric point and the presence of unstructured or intrinsically disordered extensions [66]. These features have spurred the evolution of eukaryote-specific proteins known as dedicated chaperones and escortins, which facilitate the import and/or assembly of individual r-proteins, thus impeding their degradation, inappropriate interaction with other cellular components, and aggregation [67,68,69]. Additionally, r-proteins are among the major client proteins of the general ribosome-associated chaperones such as the nascent polypeptide-associated complex (NAC) and the Ssb-RAC chaperone triad, consisting of the Hsp70 chaperone Ssb and the ribosome-associated complex (RAC) [50]. It has been shown that the loss of NAC and Ssb-RAC complexes causes aggregation of most r-proteins from both r-subunits [50] (see also Figure 8 and Figure 9). Although originally r-proteins eL40 and eS31 were not found among the proteins identified in the aggregation analysis of cells lacking NAC and Ssb-RAC (see [50]), our results show that, at least, the fused ubiquitin moiety does not prevent the specific aggregation of eS31 in the absence of the Ssb chaperone (Figure 9). Interestingly, deleting the ubiquitin moiety from the Ubi3 precursor causes a mild enhancement of the aggregation levels of eS31; however, this enhancement is not restricted to eS31 but is also observed for other directly tested r-proteins (uL29 and uS3, see Figure 9). Indeed, we show herein that a defect in ribosome biogenesis (e.g., analysis of the *ubi3∆, rps12∆, rpl39∆* or *dob1-1* mutants), similarly to what is observed in cells lacking Ssb, leads to a substantial increase in the amounts of insoluble proteins detected by polyacrylamide gel electrophoresis and Coomassie staining (Figure 8 and Figure 9). While this work was in progress, Tye et al. demonstrated that an imbalance in the synthesis of r-proteins and rRNAs leads to the rapid aggregation of newly synthesized r-proteins [57]. Thus, it is likely that the modest increase in overall aggregation detected in *ubi3∆ub* cells could also be due to the ribosome biogenesis defect caused by the *ubi3∆ub* allele, which impairs 40S r-subunit synthesis [35]. Similarly, the *UBI1*-HA *ubi2∆* strain showed a substantial amount of insoluble material and aggregation of r-proteins, as both the *ubi2∆* mutation and, to a lesser extent, the C-terminal HA-tag on Ubi1 interfere with ribosome biogenesis and cell growth [33,36] (see also Figure 2 and Figure 5A). Unfortunately, this general aggregation appears to be sufficient to mask the possible specific aggregation of eL40A-HA due to its expression from the ubiquitin-free *ubi1∆ub* allele in *ubi2∆* cells. Notably however, our results indicate that the fusion of Smt3 to eL40 seems to modestly prevent the aggregation of the processed r-protein. Altogether, these experiments, although not fully conclusive, strongly suggest that the ubiquitin moiety could indeed facilitate *in cis* the productive expression and proper folding of the fused eL40 and eS31 r-proteins.

## 5. Conclusions

The use of amenable tools in yeast biochemistry and genetics has enabled us to shed light on the biological role of ubiquitin fused to the r-proteins eL40 and eS31. Our findings provide evidence for how ubiquitin could contribute to the efficient expression and folding of its fused r-protein tails, eL40 and eS31, in order to furnish them as assembly-competent and fully functional mature r-proteins. With the exception of a ubiquitin variant that remains attached to eS31 in the protozoan parasite *G. lamblia*, there is so far no evidence of a naturally occurring non-cleaved form of ubiquitin fused to either eS31 or eL40 as a constituent of mature ribosomes [70]; thus, proteolytic removal of the ubiquitin moiety from the precursors seems to be a general requirement for the functionality of the r-subunits containing these two r-proteins. Furthermore, our results suggest that the fused ubiquitin moiety may act as an in-vivo chaperone enabling the efficient synthesis of eL40 and eS31, a function that can, although only partially, also be artificially fulfilled by the yeast ubiquitin-like Smt3 protein. In support of such a chaperone function, ubiquitin fused to eS31 has indeed evolutionarily diverged into a ubiquitin-like moiety in some organisms, such as, for example, in the nematodes *Caenorhabditis elegans* and *Caenorhabditis briggsae* [71].

## Figures and Tables

**Figure 1 cells-08-00850-f001:**
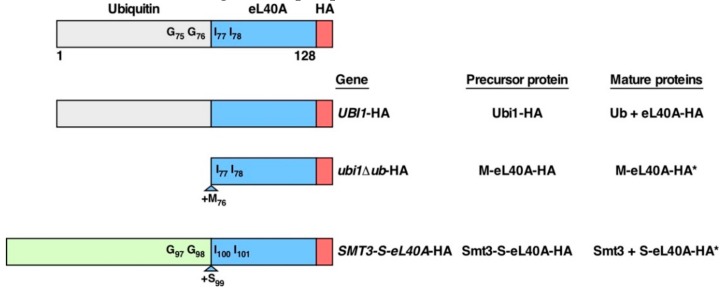
Schematic representation of the genomic constructs used in this study. Ubi1 consists of an N-terminal ubiquitin moiety fused to the r-protein eL40A. A peptide bond between the C-terminal glycine of ubiquitin (G76) and the N-terminal isoleucine of eL40A (I77) connects the two proteins. Relevant residues and *ubi1* mutant variants thereof used in this study are also indicated. The relevant constructs are schematically depicted, and the respective allele, precursor protein, and mature protein names are indicated. The asterisk denotes that the exact nature of the mature proteins derived from these precursor proteins has not been experimentally determined; thus, the indicated mature proteins correspond to the most likely in vivo cleavage products. A single C-terminal HA tag was added for western detection of the Ubi1 and/or eL40A protein variants.

**Figure 2 cells-08-00850-f002:**
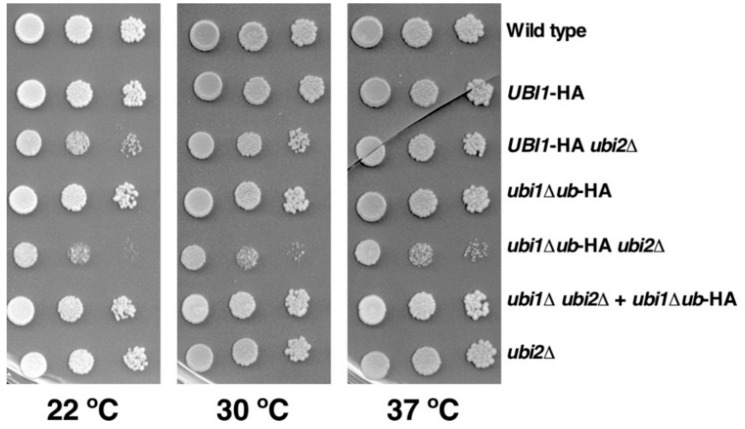
The genomically integrated *ubi1*∆*ub*-HA allele confers a slow-growth phenotype. Growth analysis of the indicated strains, either expressing a C-terminally HA-tagged Ubi1 (*UBI1*-HA) or a Ubi1∆ub (*ubi1*∆*ub*-HA) variant protein from the *UBI1* genomic locus or from a centromeric plasmid (+ *ubi1*∆*ub*-HA) in a wild-type *UBI2* or a *ubi2∆* null mutant background. Strains W303-1A (Wild type), SMY113 (*UBI1*-HA), SMY256 (*UBI1*-HA *ubi2∆*), SMY107 (*ubi1∆ub*-HA), SMY257 (*ubi1∆ub*-HA *ubi2∆*), SMY215 [YCplac111-ubi1∆ub-HA] (*ubi1∆ ubi2∆* + *ubi1∆ub*-HA), and JDY923 (*ubi2∆*) were spotted in fivefold serial dilution steps onto YPD plates, which were incubated for 4 days at 22 °C or for 3 days at 30 °C and 37 °C.

**Figure 3 cells-08-00850-f003:**
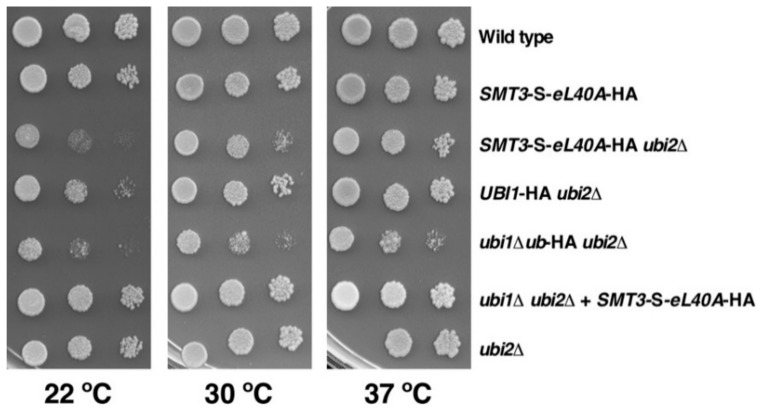
The genomically integrated *SMT3-S-eL40A*-HA allele permits better growth than the corresponding *ubi1*∆*ub*-HA allele. Growth analysis of the indicated strains, either expressing a C-terminally HA-tagged Ubi1 (*UBI1*-HA) or a Smt3-S-eL40A (*SMT3-S-eL40A*-HA) fusion variant protein from the *UBI1* genomic locus or from a centromeric plasmid (+ *SMT3-S-eL40A*-HA) in a wild-type *UBI2* or a *ubi2∆* null mutant background. Strains W303-1A (Wild type), SMY218 (*SMT3*-*S-eL40A*-HA), SMY258 (SMT3-*eL40A*-HA *ubi2∆*), SMY256 (*UBI1*-HA *ubi2∆*), SMY257 (*ubi1∆ub*-HA *ubi2∆*), SMY216 [YCplac111-SMT3-S-eL40A-HA] (*ubi1∆ ubi2∆* + SMT3-S-eL40A-HA), and JDY923 (*ubi2∆*) were spotted in fivefold serial dilution steps onto YPD plates, which were incubated for 4 days at 22 °C and for 3 days at 30 °C and 37 °C.

**Figure 4 cells-08-00850-f004:**
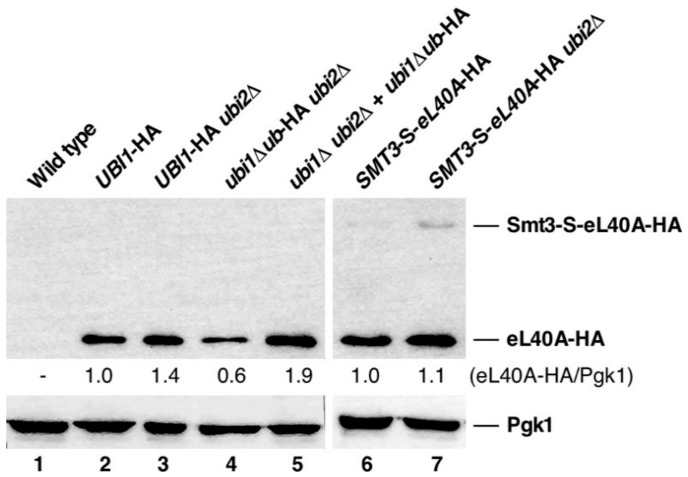
Analysis of the protein levels of eL40A-HA. A wild-type control strain (lane 1) and strains with the indicated relevant genotypes, expressing C-terminally HA-tagged eL40A from the different precursor constructs, either from the genomic locus (lanes **2**, **3**, **4**, **6**, and **7**) or from a centromeric plasmid (lane **5**), in a wild type *UBI2* or a *ubi2∆* null mutant background, were grown in liquid YPD medium at 30 °C. Total cell extracts were prepared and subjected to western analysis using anti-HA and anti-Pgk1 (loading control) antibodies. The mature eL40A-HA and the Smt3-S-eL40A-HA precursor are indicated.

**Figure 5 cells-08-00850-f005:**
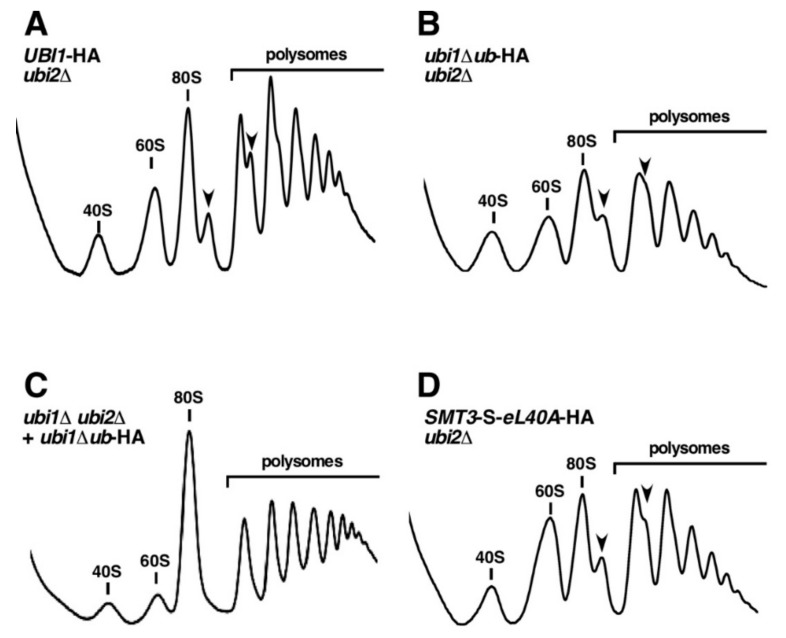
The genomically integrated *ubi1*∆*ub*-HA and *SMT3-S-eL40A*-HA alleles affect the synthesis and function of 60S r-subunits. Strains SMY256 (*UBI1*-HA *ubi2∆*) (**A**), SMY257 (*ubi1∆ub*-HA *ubi2∆*) (**B**), SMY215 [YCplac111-*ubi1∆ub*-HA] (*ubi1∆ ubi2∆ + ubi1∆ub*-HA) (**C**), and SMY258 (*SMT3-S-eL40A*-HA *ubi2∆*) (**D**) were grown in YPD medium at 30 °C to an OD_600_ of about 0.8. Cell extracts were prepared and 10 A_260_ units of each extract were resolved in 7–50% sucrose gradients. The A_254_ was continuously measured. Sedimentation is from left to right. The peaks of free 40S and 60S r-subunits, vacant 80S ribosomes/monosomes and polysomes are indicated. Half-mer polysomes are labeled by arrows.

**Figure 6 cells-08-00850-f006:**
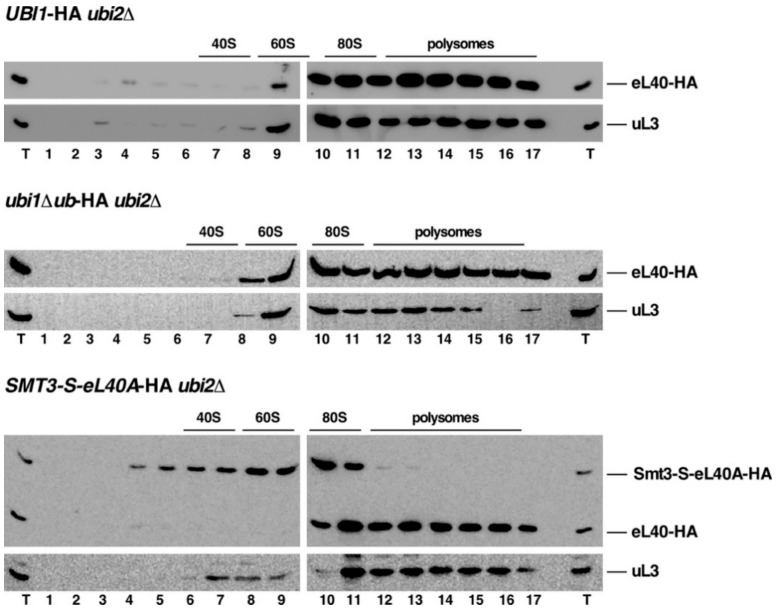
Incorporation of eL40A-HA, derived from the Ubi1-HA, Ubi1∆ub-HA, and Smt3-S-eL40A-HA precursors, into translating ribosomes. Strains SMY256 (*UBI1*-HA *ubi2∆*), SMY257 (*ubi1∆ub*-HA *ubi2∆*), and SMY258 (*SMT3-S-eL40A*-HA *ubi2∆*) were grown in YPD medium at 30 °C to mid-log phase. Cell extracts were prepared and 10 A_260_ units of each extract were resolved in 7–50% sucrose gradients. Fractions were collected from the gradients, the proteins were extracted from each fraction, and equal volumes were analyzed by western blotting using anti-HA and anti-uL3 antibodies. The position of free 40S and 60S r-subunits, vacant 80S ribosomes/monosomes, and polysomes, obtained from the recorded A_254_ profiles, are shown. T, total cell extract.

**Figure 7 cells-08-00850-f007:**
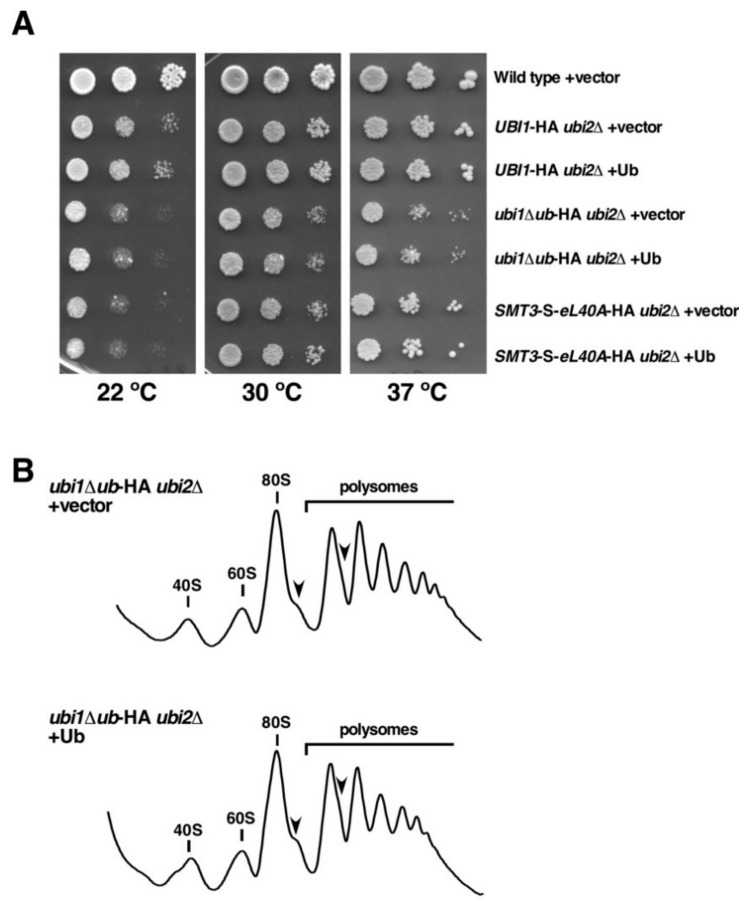
Expression of free ubiquitin *in trans* does not restore the growth defects of *ubi1∆ub*-HA *ubi2∆* or *SMT3-S-eL40A*-HA *ubi2∆* cells. (**A**) Growth comparison of the strains W303-1A (Wild type), SMY256 (*UBI1*-HA *ubi2∆*), SMY257 (*ubi1∆ub*-HA *ubi2∆*), and SMY258 (*SMT3*-*S*-*eL40A*-HA *ubi2∆*) transformed with either empty YCplac111 (vector) or pADH111-Ub (Ub), a YCplac111-based plasmid expressing ubiquitin from the strong *ADH1* promoter. Transformants were selected on SD-Leu plates and then spotted in fivefold serial dilution steps onto SD-Leu plates, which were incubated for 4 days at 22 °C, for 3 days at 30 °C, and for 2 days at 37 °C. (**B**) Polysome profile analysis of *ubi1∆ub*-HA *ubi2∆* cells harboring either empty YCplac111 (vector) or pADH111-Ub (Ub). Cells were grown in SD-Leu at 30 °C to an OD_600_ of about 0.8. Cell extracts were prepared and 10 A_260_ units of each extract were resolved in 7–50% sucrose gradients. The A_254_ was continuously measured. Sedimentation is from left to right. The peaks of free 40S and 60S r-subunits, vacant 80S ribosomes/monosomes, and polysomes are indicated. Half-mer polysomes are labeled by arrows.

**Figure 8 cells-08-00850-f008:**
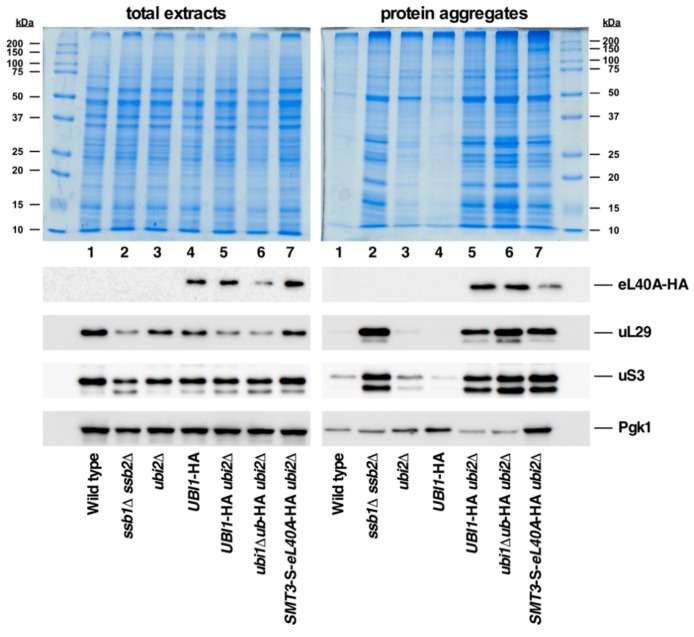
Analysis of protein aggregation in *ubi1∆ub*-HA and *SMT3-S-eL40A*-HA cells. Strains W303-1A (Wild type), JDY532 (*ssb1∆ ssb2∆*), JDY923 (*ubi2∆*), SMY113 (*UBI1*-HA), SMY256 (*UBI1*-HA *ubi2∆*), SMY257 (*ubi1∆ub*-HA *ubi2∆*), and SMY258 (*SMT3*-*S*-*eL40A*-HA *ubi2∆*) were grown to logarithmic phase in YPD medium at 30 °C. Then, total protein extracts and protein aggregates were prepared, separated by SDS-PAGE, and visualized by Coomassie staining (upper part) or subjected to western blot analysis (lower part) using antibodies detecting either the HA-tag of the eL40-HA protein or the uL29, uS3, and Pgk1 proteins.

**Figure 9 cells-08-00850-f009:**
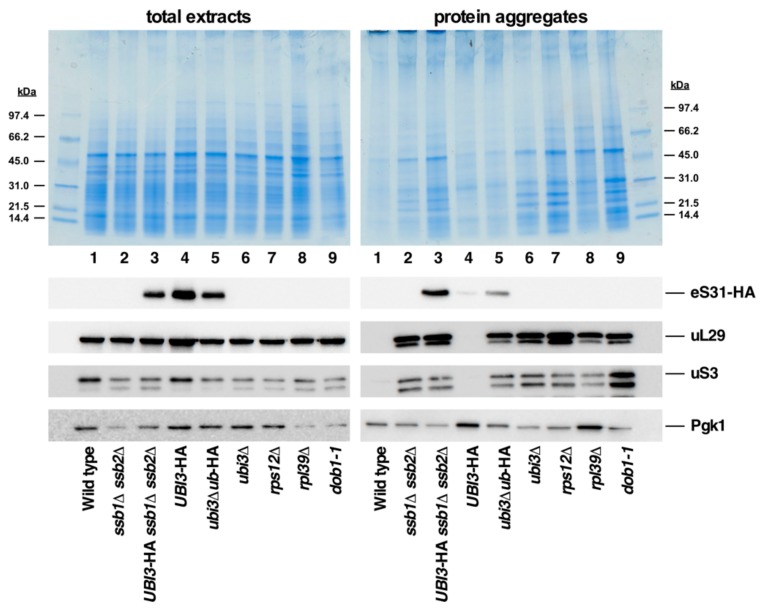
Analysis of protein aggregation in the *ubi3∆ub*-HA strain and cells lacking individual r-proteins or being defective in ribosome biogenesis. Strains W303-1A (Wild type), JDY532 (*ssb1∆ ssb2∆*), SMY324 (*UBI3*-HA *ssb1∆ ssb2∆*), TLY56.D3 (*UBI3*-HA), TLY61.A2 (*ubi3∆ub*-HA), TLY14.3C (*ubi3∆*), SMY315 (*rps12∆*), ORY211 (*rpl39∆*), and MS157-1A (*dob1-1*) were grown to logarithmic phase in YPD medium at 30 °C. Then, total protein extracts and protein aggregates were prepared, separated by SDS-PAGE, and stained with Coomassie (upper part) or subjected to western blot analysis (lower part) using anti-HA (detection of eS31-HA), anti-uL29, anti-uS3, and anti-Pgk1 antibodies.

**Table 1 cells-08-00850-t001:** Yeast strains used in this study.

Strain ^a^	Relevant Genotype ^b^	Source
W303-1A	*MAT* **a** *ade2-1 his3-11,15 leu2-3,112 trp1-1 ura3-1*	[37]
W303-1B	As W303-1A but *MAT*α	[37]
TAY001	*MAT***a***ubi1*::*kanMX4* ubi2::kanMX4 *ade3*::kanMX4 [pHT4467∆-UBI1]	[36]
SMY106	*MAT***a***ubi1*::klURA3	This study
SMY113	*MAT*a *UBI1*-HA	This study
SMY256	*MAT***a***UBI1*-HA *ubi2*::kanMX4	This study
SMY107	*MAT***a***ubi1∆ub*-HA	This study
SMY257	*MAT*α *ubi1∆ub*-HA *ubi2*::kanMX4	This study
SMY215	*MAT***a***ubi1*::kanMX4 *ubi2*::kanMX4 *ade3*::kanMX4 [YCplac111-ubi1∆ub-HA]	This study
JDY923	*MAT*α *ubi2*::kanMX4	[33]
SMY218	*MAT***a***SMT3*-S-*eL40A*-HA	This study
SMY258	*MAT*α *SMT3-S-eL40A*-HA *ubi2*::kanMX4	This study
SMY216	*MAT***a***ubi1*::kanMX4 *ubi2*::kanMX4 *ade3*::kanMX4 [YCplac111-SMT3-S-eL40A-HA]	This study
JDY532	*MAT***a***ssb1*::HIS3MX6 *ssb2*::natNT2	This study
SMY324	*MAT***a***ssb1*::HIS3MX6 *ssb2*::natNT2 *UBI3*-HA::kanMX4	This study
TLY56.D3	*MAT***a***UBI3*-HA::kanMX4	[35]
TLY61.A2	*MAT***a***ubi3∆ub*-HA::kanMX4	[35]
TLY14.3C	*MAT*α *ubi3*::HIS3MX6	[35]
SMY315	*MAT*α *rps12*::kanMX4	This study
ORY211	*MAT*α *rpl39*::natNT2	This study
MS157-1A	*MAT*α *dob1-1*	[42]

**^a^** Strains used in this study are isogenic with W303. **^b^** For simplicity, only the relevant genotypes, differing from the one of the W303-1A strain, are indicated.

**Table 2 cells-08-00850-t002:** Plasmids used in this study.

Name (Collection Name)	Relevant Information	Source
pHT4467∆-UBI1	*CEN6* (instable), *URA3*, *ADE3*. Wild-type Ubi1; promoter and terminator of *UBI1*	[36]
YCplac111	*CEN*, *LEU2*	[43]
YCplac111-UBI1-HA (pDK4131)	*CEN*, *LEU2*. C-terminally 1xHA-tagged eL40A; promoter and terminator of *UBI1*	[36]
YCplac111-ubi1∆ub-HA (pDK4192)	*CEN*, *LEU2*. Allele *ubi1∆ub*-HA (M-II); promoter and terminator of *UBI1*	This study
YCplac111-SMT3-S-eL40-HA (pDK4193)	*CEN*, *LEU2*. Allele *SMT3*-*S*-*eL40A*-HA (GG-S-II); promoter and terminator of *UBI1*	This study
pADH111-Ub (pDK2253)	*CEN*, *LEU2*. Wild-type ubiquitin; promoter and terminator of *ADH1*	This study

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
