# Peer review of "The Ubiquitin Moiety of Ubi1 Is Required for Productive Expression of Ribosomal Protein eL40 in Saccharomyces cerevisiae"

_cells, 2019, doi:10.3390/cells8080850_

Round 1
Reviewer 1 Report
In this manuscript Martin-Villanueva et al use Yeast genetics and cell-based assays to further our understanding of the role of the Ubi1 gene in ribosome production. The authors created several yeast strains to see what effect removal or replacement of the ubiquitin moiety fused to eL40 has on yeast viability, ribosome assembly, and L40 expression and solubility. The authors show that ubiquitin is required for optimal expression of L40 and that its removal causes growth defects, a decrease in L40 protein levels in vivo, and defects in ribosome production. Moreover, they demonstrate that the decrease in L40 protein levels due to the removal of ubiquitin can be rescued by addition of a SUMO moiety. Overall this is a well-written manuscript that provides insight into a long-standing question about the function of the ubiquitin moiety fused to eL40. Once the authors address the concerns below (which are mostly minor) then I think this manuscript should be published in Cells.
Concerns
1. The authors should merge Figures 5 and 6, since this is essentially the same experiment. It would be easier to follow if the Western Blots were shown beneath the corresponding sucrose gradient profiles.
2. Based on the Western blots in Figure 6 the authors suggest that failure to remove the SUMO tag from L40 blocks formation of active polysomes. This argument would be better supported with a non-cleavable Smt3-L40A-HA variant.
3. More information is needed about the isolation of aggregated proteins as it is currently unclear how these experiments were done. Please provide more experimental details so that readers can understand the experiment without having to look up the references.
4. Why is Pgk1 a suitable Western blot control in Figures 8 and 9?
5. This manuscript is missing any qualifiers on reproducibility. Please include the number of biological and/or technical replicates performed in each figure legend.
Author Response
Reply to the review report (Reviewer 1)
The authors should merge Figures 5 and 6, since this is essentially the same experiment. It would be easier to follow if the Western Blots were shown beneath the corresponding sucrose gradient profiles.We completely understand the suggestion of this reviewer; however, we prefer not to merge the two Figures. The profiles shown in Figure 5 do not correspond to the same biological and technical replicates as the ones used for the fractionation and subsequent western blotting analysis shown in Figure 6. Moreover, our Figure 6 does not show the fractionation analysis of the strain whose profile is shown in Figure 5C, as this profile was indistinguishable of that obtained for a wild-type strain. Thus, the merged Figure would give the impression of being incomplete. We hope that you agree with our arguments.
Based on the Western blots in Figure 6 the authors suggest that failure to remove the SUMO tag from L40 blocks formation of active polysomes. This argument would be better supported with a non-cleavable Smt3-L40A-HA variant.
This is a good suggestion. However, non-cleaved Smt3-L40A-HA variants are not viable, even when expressed from plasmids. Similar results were previously obtained for non-cleaved Ubi1 mutants. As occurring for the Smt3-L40A-HA precursor (Figure 6), the non-cleaved Ubi1 precursors got incorporated into 60S and 80S peaks but were practically absent from the polysomal fractions (please, see our recent publication by Martín-Villanueva et al. FEBS J. 2019; doi: 10.1111/febs. 149999).
More information is needed about the isolation of aggregated proteins as it is currently unclear how these experiments were done. Please provide more experimental details so that readers can understand the experiment without having to look up the references.
We have now provided further details on the experimental procedure of the isolation of protein aggregates. Please, see the current section 2.5 of Materials and Methods.
4.Why is Pgk1 a suitable Western blot control in Figures 8 and 9?
Pgk1 (phosphoglycerate kinase) is a housekeeping enzyme involved in glycolysis and gluconeogenesis. It is apparently not involved in any aspect of ribosome metabolism. Housekeeping enzymes are commonly used loading controls. Moreover, as shown in Figure 8, Pgk1 has a low propensity for aggregation, even in the absence of the ribosome-associated Ssb-RAC chaperone system. These explanations have now been incorporated into the text of the revised manuscript (please, see revised Results section 3.7, page 17, lines 483-485).
This manuscript is missing any qualifiers on reproducibility. Please include the number of biological and/or technical replicates performed in each figure legend.
In our laboratory, we usually perform three biological replicates for each experiment and the same samples were analyzed at least two times (technical replicates). We have included a new section (section 2.6) in Material and Methods named "Reproducibility" with that statement, which applies for all the Figures of our revised manuscript.
Reviewer 2 Report
The manuscript by Martin-Villanueva et al describes novel functions of the fused ubiquitin moiety which might serve as a molecular chaperone. The authors previously described the removal of ubiquitin moiety from ribosomal protein eL40 is essential for both the cytoplasmic maturation and the functionality of 60S ribosomal subunits (FEBS J, 2019). The authors have now added the biological role of ubiquitin fused to the ribosomal proteins. The study is performed to a very high technical standard and contains many elegant experiments. However, the major criticism is the current paucity of data that would describe a strong link between the slow-growth phenotype and protein aggregate.
As shown in Fig.8, protein aggregation was not found in wild type, ubi2Δ, and UBI1-HA. By contrast, protein aggregation was detected in UBI1-HA ubi2Δ. How the authors reconcile these observations? Was protein aggregation detected in ubi1Δub-HA? Slow-growth phenotype and protein aggregates may not be related. Besides, it is difficult to distinguish the smear intensity difference between UBI1-HA ubi2Δand ubi1Δub-HA ubi2Δ in protein aggregates by Coomassie staining in Fig.8.
The authors mentioned that protein aggregates were observed in ubi3Δub-HA in Fig.9. However, the blue smear was only faint in ubi3Δub-HA in protein aggregates by Coomassie staining. How do the authors think about these observations? How were the growth phenotype in UBI3-HA and ubi3Δub-HA?
Another comment;
It is easy to understand the data if the authors add lane numbers in Fig.8 and Fig.9. Please add lane numbers in the text, too.
Author Response
Reply to the review report (Reviewer 2)
The major criticism is the current paucity of data that would describe a strong link between the slow-growth phenotype and protein aggregate.
We understand the concern of the reviewer; however, we do not claim at any point of the manuscript that the slow-growth phenotypes of the studied strains are due to the observed protein aggregation status.
As shown in Fig.8, protein aggregation was not found in wild type, ubi2Δ, and UBI1-HA. By contrast, protein aggregation was detected in UBI1-HA ubi2Δ. How the authors reconcile these observations?
These are indeed puzzling observations. We firmly believe that the observed aggregation in UBI1-HA ubi2∆ cells is due to the combined effects of the absence of Ubi2, which leads to a slight ribosome production defect (see our publication by Fernández-Pevida et al. J. Biol. Chem. 287: 38390-38407, 2012), and the presence of the C-terminal HA tag on L40A. Indeed, this strain is growing slower than the isogenic UBI1-HA and ubi2∆ strains (see Figure 2). We had previously mentioned this argument in our initial manuscript, which has been kept in our revised version (please, see Discussion section, page 22, lines 631-634).
Was protein aggregation detected in ubi1Δub-HA?
As the growth of a ubi1∆ub-HA strain is apparently indistinguishable from that of the UBI1-HA strain (see Figure 2), we have never tested the protein aggregation status in ubi1∆ub-HA cells.
Slow-growth phenotype and protein aggregates may not be related. Besides, it is difficult to distinguish the smear intensity difference between UBI1-HA ubi2Δand ubi1Δub-HA ubi2Δin protein aggregates by Coomassie staining in Fig.8.
As we have answered above, we do not state a correlation between the slow-growth and protein aggregation phenotypes. Moreover, we agree with this reviewer that it is difficult to distinguish the smear intensity differences in protein aggregates between strains by Coomassie staining, thus is why we have only carefully mentioned in the text that "the global aggregation tendency was very similar in the three strains (UBI1-HA ubi2∆, ubi1∆ub-HA ubi2∆ and SMT3-S-eL40A-HA), albeit slightly more pronounced in ubi1∆ub-HA ubi2∆ cells".
The authors mentioned that protein aggregates were observed in ubi3Δub-HA in Fig.9. However, the blue smear was only faint in ubi3Δub-HA in protein aggregates by Coomassie staining. How do the authors think about these observations?
We agree with the reviewer that the global aggregation status of the ubi3∆ub-HA mutant seems not to be enhanced versus that of the isogenic UBi3-HA when assessed by colloidal blue Coomassie staining. This observation gives even more value to the fact that eS31-HA and other ribosomal proteins are clearly found in the aggregates.
How were the growth phenotype in UBI3-HA and ubi3Δub-HA?
The ubi3∆ub-HA mutant has a severe slow-growth phenotype (see Discussion, page 20, line 553-555) that it is less dramatic than the one of the ubi3∆ null mutant. We have previously published these results (see Lacombe et al. Mol. Microbiol. 72: 69-84, 2009).
It is easy to understand the data if the authors add lane numbers in Fig.8 and Fig.9. Please add lane numbers in the text, too.
We have followed this advice (please, see current Figures 8 and 9). To keep the text easily readable, we have only included the lane numbers in the text in the most important cases (please, see revised Results section 3.8).